# On the Technological Acceptance of Moodle by Higher Education Faculty—A Nationwide Study Based on UTAUT2

**DOI:** 10.3390/bs13050419

**Published:** 2023-05-15

**Authors:** Gabriel García-Murillo, Pavel Novoa-Hernández, Rocío Serrano Rodríguez

**Affiliations:** 1Faculty of Philosophy, Literature and Education Sciences, Universidad Técnica de Manabí, Portoviejo 130105, Ecuador; 2Models of Decision and Optimization Research Group, University of Granada, 18014 Granada, Spain; 3Faculty of Education Sciences and Psychology, Department of Education, University of Cordoba, 14071 Cordoba, Spain

**Keywords:** Moodle, learning management systems, higher education, academic staff, utaut2, PLS-SEM

## Abstract

Moodle is an open-source learning management system that is widely used today, especially in higher education settings. Although its technological acceptance by undergraduate students has been extensively studied in the past, very little is known about its acceptance by university professors. In particular, as far as we know, the literature contains no previous experiences related to South American teachers. This paper aims to bridge this gap by quantifying and analyzing the drivers of Moodle’s technological acceptance among Ecuadorian academic staff. Considering the responses of 538 teachers and taking a modified UTAUT2 model as a theoretical basis, we found that Ecuadorian teachers have high levels of acceptance of Moodle, regardless of their age, gender, ethnicity, or discipline. However, this acceptance is significantly higher in teachers with high levels of education and with considerable previous experience with e-learning systems. The main determinants of this acceptance are attitude strength, effort expectancy, performance expectancy, and facilitating conditions. We found no moderating effects in relation to the age, gender, or previous experience of the participants (including second- and third-order interactions derived from these variables). We conclude that, albeit moderately (e.g., adjusted $R^{2}=0.588$), the model tested confirms the predictive power of the part of UTAUT2 that was inherited from UTAUT.

## 1. Introduction

The impact of information and communication technologies (ICT) on society at the present time is undeniable [1]. In education, ICT has contributed to the exploration of new ways of teaching and learning, which have proven particularly effective in an increasingly connected and globalized world [2,3]. Most of these approaches, such as learning management systems (LMS), are based on software tools that enhance and manage learning [4].

As in other software families, LMSs offer both paid and free alternatives. *Blackboard* (www.blackboard.com) and *Desire2Learn* (www.d2l.com) are popular examples of the first group, while *Moodle* (www.moodle.org) and *Claroline* (www.claroline.net) are considerably popular examples of the second. Moodle, originally created in 2001 and currently on version 3.11, has not only significantly fulfilled its initial purpose (i.e., to enable the efficient management of online learning) but has also created an online collaboration community that allows the concept of LMS to evolve, while taking into account the most consumed learning approaches and technologies in society [5].

Distributed under the GNU General Public License (as published by the Free Software Foundation) [6], Moodle is undoubtedly one of the most widely used LMSs today [7]. According to the statistics reported by stats.moodle.org, more than 200 million users from 244 countries use Moodle. Much of this popularity is owing to the fact that it is open-source, and can therefore be adapted to the most diverse of scenarios [8].

Although LMSs, particularly Moodle, have been extensively studied in the field of technological acceptance [9,10,11], several issues remain, such as technological acceptance by university teachers. According to García-Murillo et al. [12], despite a considerable number of publications pertaining to university students, including preservice teachers, faculty members remain understudied. In our opinion, given the fact that teachers form an essential part of the learning process, the extent to which they accept and use these technologies should be understood. This perception is consistent with [13], who found that the more actively educators use Moodle, the more actively students tend to use it too. More active use by students effectively translates into meaningful learning [14].

This paper seeks to contribute to closing the gap in existing research on this subject by studying the technological acceptance of Moodle by higher education faculty in Ecuador. The main objective of our work is to characterize this technological acceptance and identify its most significant determinants. To our knowledge, there is no previous work of research that touches upon this issue in relation to South America, particularly Ecuador, which is one of the countries with a high adoption rate of Moodle by higher education institutions (According to the website https://stats.moodle.org/sites/, Ecuador has more than 3400 registered sites). Consequently, the results of our research seek to advance research on the technological acceptance of LMSs in higher education environments.

## 2. Background and Related Work

The acceptance of information systems has been extensively studied in the past under technology acceptance models [15]. Several approaches have been employed, beginning with the seminal work of Davis [16] with the very popular technology acceptance model (TAM), which is based on the theory of planned behavior (TPB) by Ajzen [17]. Consequently, the actual system use (ASU) is conceived here as a behavior, while the perceived ease of use (PEU), perceived usefulness (PU), and attitude toward use (ATU) are the determinants of such a behavior [16]. More specifically, this model states that the user’s attitude toward the system is crucial in determining whether or not the user will actually employ the system. *PU* is conceived as the degree to which an individual believes that using that technological system will improve his or her performance, while *PEU* is defined as the degree to which the individual believes that using that particular technological system does not require extra effort or skills. TAM assumes that both beliefs are directly influenced by the design characteristics of the system and external variables. In further refinements of TAM, the behavioral intention (BI) to use was considered a determining factor of ASU and, at the same time, a dependent factor of *ATU* [18]. Later, in [19], it was found that both PU and PEU have direct effects on BI, and it was therefore not necessary to rely on *ATU*. TAM2 Venkatesh and Davis [20] and TAM3 [21] are two other important extensions of the original TAM. They involve external variables that aim to explain the user’s *PU* and *PEU*. Such variables were grouped into the following four categories [21]: individual differences, system characteristics, social influence, and facilitating conditions.

Regardless of the significant impact of TAM—the theory of technological acceptance—some authors proposed other alternatives. Perhaps the most popular is the unified theory of acceptance and use of technology (UTAUT), proposed by Venkatesh et al. [22]. Following a different approach than TAM2 and TAM3, UTAUT replaces *PU* and *PEU* with four determinants: *performance expectancy* (PE), *effort expectancy* (EE), *social influence* (SI), and *facility conditions* (FC). Thus, these new determinants are expected to explain BI, which, in turn, should determine the use behavior (UB). UTAUT also hypothesizes that most of these relationships are moderated by the users’ age (AGE), gender (GDR), experience (EXP), and voluntariness of use (VOL). UTAUT was updated by Venkatesh et al. [23] to the proposed model known as UTAUT2. The authors added the three following new determinants for BI: *hedonic motivation* (HM), *price value* (PV), and *habit* (HT). It should be noted that in UTAUT2, the variable VOL was excluded as a moderator of the relationships predicting BI, while AGE, GDR, and EXP were maintained as moderating variables (including the variables formed from the second- and third-order interactions between them). More details on the specific definition of each construct included in UTAUT and UTAUT2 can be found in Table A1 (Appendix A).

As recently shown by Murillo et al. [10], García-Murillo et al. [12], Moodle acceptance has been the focus of several studies over the past 15 years. Most of these studies were oriented to characterize the technological acceptance of university students from Europe or Asia and to employ TAM as the base model. In the specific case of academic staff, the few available experiences reported good levels of acceptance of Moodle in general [11,24].

For instance, Costa et al. [25] reported that 96 professors of the University of Aveiro in Portugal accepted Moodle despite not being familiar with the MOOC concept [26]. This study also found only a few significant differences among the respondents’ gender, knowledge area, and age. Social influence (SI) was found to be a determinant of PU but not of PEU. Moreover, the authors confirmed the significant effects of PU and PEU on ATU. Taking into account the gender of the respondents, the fitted model shows that PEU had no significant effects on PU in the case of male participants, while it did in the case of females. Similarly, for respondents related to social science and humanities, PEU was a significant determinant of PU, while for the rest of the knowledge areas, it was not.

In the same line, but relying on a modified UTAUT model, Islam [27] explored the determinants of the professors’ continuing intention of using Moodle. He found that it is explained by PU and access (A), while PU is predicted by PEU and compatibility (C). Overall, 70% of the continuance intention is explained by the six variables considered. The study was based on 175 college professors from a Finnish university.

The perceptions of the professors toward Moodle were evaluated by Baytiyeh [28] using the UTAUT model. To this end, data coming from 189 respondents at a Lebanese university were analyzed following exploratory factor analysis [29] and a multiple regression approach [30]. The first analysis enabled the identification of the following five factors: community influence (CI), satisfaction (S), service quality (SQ), learnability (L), and technical quality (TQ). The second one showed that these five factors significantly influence system use (SU), which was assumed to be the core variable related to the technological acceptance of Moodle.

Motivated to explore whether students’ perceptions of learning technologies are different from those of professors, North-Samardzic and Jiang [31] relied on the UTAUT model. Overall, some similarities and differences were identified. In the case of professors, it was found that *EE* was the most important factor to explain *BI*. However, *FC* did not influence *UB*, while Age had a direct effect on *UB* as a moderator. Due to the lower number of accepted hypotheses, the authors suggested that UTAUT, at least in its original form, may not be the right model to study technology acceptance in higher education settings. This study based its findings on a sample size of 89 professors at an Australian university.

Another interesting study was conducted by Zwain [32], in which UTAUT2 was expanded by considering two new predictors. The first, technological innovativeness (TI), aims to measure the degree of readiness in using a new technology [33], while the second, information quality (IQ), is devoted to capturing the perceived quality of the information provided by the system [34]. Another important modification made to the original model was the employment of learning value (LV) as a more realistic alternative to price value [35]. The rationale behind this modification stems from the fact that Moodle is an open-source software, and hence, the final users are not expected to perceive any economic benefits from it. Since these final users come from educational settings, Ain et al. [35] suggested replacing *PV* with *LV*. A structural equation modeling (SEM) approach was adopted in the study using 228 responses from professors at the University of Kufa, Iraq. As a result, the author found that *SI*, *FC*, *HM*, *HT*, *TI*, and *IQ* significantly determine Moodle’s acceptance in terms of BI and UB. More recently, Karkar et al. [36] adopted a data mining approach to highlight the major challenges while adopting Moodle in the same university. The authors found from 242 professors that they find social media platforms easier to use than Moodle.

The attitudes of 199 B-school professors from India were analyzed by Kushwaha et al. [37] using TAM as the underlying theory. Here, both PEU and PU were identified as significant predictors of ATU, while certain demographic features, such as home city, gender, and age, were highlighted as significant moderators of some of these relationships.

Regardless of the progress made by the studies discussed above, it is clear that much more work remains to be carried out to understand how academic staff accept Moodle. As a general pattern, we can observe that the reported experiences are scarce and heterogeneous in both results and models tested. Moreover, they are generally based on a few single-institution respondents and are carried out in specific regions (e.g., Asia or Europe). Another characteristic of these studies is that the predominant base model has been UTAUT (including its most recent extension, UTAUT2).

## 3. Research Questions and the Hypothesized Model

Based on the above rationale and background, this study seeks to answer the following research questions:*RQ1) What is the level of technological acceptance of Moodle by Ecuadorian higher education professors?* In answering this question, we sought to quantify the level of acceptance by Ecuadorian higher education professors, specifically in the context of blended learning, which occurred during the period of social isolation brought on by the COVID-19 pandemic. For this purpose, following Nistor et al. [38], we assumed *BI* as the construct that will capture the users’ acceptance of Moodle. We want to verify whether the proportion of professors accepting Moodle is similar to the one reported by Garcia-Murillo et al. [11], which was approximately 0.69 with a 95% confidence interval [0.59,0.78].A relevant question associated with the previous one is this:–*RQ1a) Is the technological acceptance of Moodle the same regardless of professor demographics?* With this question, we seek to explore whether certain demographic characteristics of teachers are associated with specific levels of technological acceptance. Previous studies, such as North-Samardzic and Jiang [31], Kushwaha et al. [37], focused on analyzing the extent to which some of these variables (such as gender and age, among others) moderate the relationships of the exogenous variables with the constructs that represent the technological acceptance of Moodle. However, very little evidence currently exists on the analysis of differences between groups in relation to the technological acceptance of Moodle. So, by investigating this issue, we will be contributing to bridging this gap.*RQ2) What are the determinants of Moodle’s acceptance by Ecuadorian higher education professors?* The purpose of this question is to identify the factors that significantly influence Moodle’s level of acceptance. In this context, given the previous experiences [27,28,31,32], we also considered relying on the UTAUT2 model. Specifically, we hypothesize that the acceptance of Moodle by Ecuadorian teachers can be explained by the model illustrated in Figure 1.In addition to UTAUT2 factors, we considered three other factors previously studied in the literature. The first one, named *attitude strength* (*AS*), was initially studied by Nistor et al. [38] in the context of university students, and was defined as “the degree to which attitude manifests itself in the form of temporal persistence, resistance to counter persuasion and predictability of behavior” ([38], p. 4). In that study, the authors hypothesized significant direct effects of *AS* on *PE*, *EE*, *SI*, and *FC*. Such hypotheses were confirmed only for the first three factors, but not for *FC*, due to the poor reliability of this construct. Therefore, it is possible that *AS* could also influence *FC* if it were measured with sufficient reliability and validity [39]. This is a hypothesis that we intend to test in the context of university teachers.The remaining determinants were *learning value* (LV) and *technological innovativeness* (TI). As noted above, the former is a redefinition of price value proposed by Ain et al. [35], while the latter was investigated by Zwain [32]. In both cases, the authors found significant effects of these constructs on *BI*. The other construct investigated by Zwain [32], called IQ, was discarded because, in our opinion, the purpose of Moodle is not to provide quality information to teachers but to enable them to manage their teaching practice. In addition, we reformulated the *LV* construct to better adapt it to the teaching context. In this case, we found it better to call it *teaching value* (TV) instead of *learning value*.It is important to note that other constructs could have been considered as well. However, our approach has been more confirmatory than exploratory from the evidence reported in the literature. In other words, we are interested in investigating the extent to which the theory that has explained the technological acceptance of Moodle in university teachers fits the Ecuadorian context.Finally, as suggested by Venkatesh et al. [23], we included in our analysis the moderating effect of background variables such as age, gender, and previous experience with LMSs. Specifically, we studied the individual and combined effects of these variables, as shown in Figure 1.

In summary, our research tested a total of 66 hypotheses, depicted in Figure 1, where 12 were direct effects (e.g., arrows going out of *AS* (4) and arrows going into *BI* (8)), 1 was an indirect effect (*AS* toward *BI*), and 53 were moderating effects related to age, gender, and experience.

Table 1 more clearly specifies which hypotheses we will be testing in this research. Note that in the first column appear the direct and indirect effects, while those related to the moderation of these effects appear in the columns corresponding to the variables *age*, *gender*, *experience*, and the combination of them (e.g., indicated by &). For example, hypothesis *H6* is read as follows: *PE has a direct effect on BI*; while *H6.1* is read as the following: *the direct effect of PE on BI is moderated by age*. The rest of the hypotheses are formulated (and interpreted) in a similar way.

## 4. Methodology

This study adopted a quantitative approach that assumes that the variables under study are latent. More specifically, a correlation design based on cross-sectional data was employed [40].

### 4.1. Population and Sample

The population to be characterized consisted of all university professors who use Moodle in Ecuador. Therefore, it was necessary to first identify which higher education institutions (HEIs) were officially using Moodle as an LMS. In this regard, we initially relied on two main sources: (1) the official Moodle site, which records installations by country; and (2) phone calls made to institutions that did not appear in the first source. As a result, of the country’s 60 HEIs, 42 were found to officially use Moodle (i.e., 70%). Based on the official statistics published by the Higher Education Council of Ecuador (https://www.ces.gob.ec/), these 42 institutions represent a population of N=43,227 academics. In this sense, according to Ryan [41], the sample required for this population in order to achieve a 95% confidence level and a 5% margin of error is n=384. Based on this sample size and the proportion of faculty per institution, it was possible to specify a stratified sample. Thus, the composition of the final sample was representative at the national level.

### 4.2. Measures and Instruments

A questionnaire was designed to collect the data for this study. It comprised 39 questions, 7 of which concerned demographics and the rest of which (32) were related to specific constructs of the model under study (Table A1, Appendix A). As considered by Nistor et al. [38], we assume *BI* to be the construct that characterizes the technological acceptance of Moodle.

The variables of the model to be studied were measured through indicators that were evaluated by the respondents using a 5-point *Likert* scale, with the following meanings: 1=
*totally disagree*, 2=
*disagree*, 3=
*neither disagree nor agree*, 4=
*agree*, and 5= *totally agree*.

The Spanish version of the questionnaire was reviewed by 6 experts in the field, 16 of whom were invited by mail. The suggestions made by the experts were included in a new version that was piloted with the faculty members of the Universidad Técnica de Manabí from Ecuador (one of the higher education institutions participating in the study). A total of 20 faculty members answered the questionnaire in person. On average, participants took 8 min to complete the questionnaire. The suggestions made by the respondents were included in a new version of the questionnaire, which was the one we administered on a national scale.

### 4.3. Data Collection

Based on the target sample size (n=384) and the presumption of a 10% response rate to the questionnaire, 3840 invitations were sent to higher education institutions in October 2020. After two months, we received 552 responses, of which 14 were not completed, i.e., respondents chose not to participate in the study. Therefore, the final sample was n=538, with sufficient responses for each HEI considered by stratified sampling. No *outliers* were identified. The main characteristics of these 538 respondents are presented in Table 2.

From Table 2, we can observe that the majority of the respondents are male, between 35 and 54 years, and self-identify as *Mestizo*. Additionally, they hold master’s degrees, with careers in social sciences, have a self-perception that their experience with LMSs is moderate, and own a computer.

### 4.4. Data Analysis

To answer research question 1 (Section 3), we relied on descriptive statistics obtained from the data collected. Specifically, we averaged the indicators that define the *BI* construct to obtain the technological acceptance of each respondent.

Research question 1.a was by conducting nonparametric statistical tests and taking technological acceptance as the main variable (e.g., as computed for research question 1). The main reason for this was the disparity between the subsample sizes generated by the levels of the demographic variables. As a consequence, it was difficult to ensure normality in the data. In the case of variables with only two levels (e.g., *Gender*), we relied on the Mann–Whitney U test for unpaired samples [42]. For variables with more than two levels, such as *Age group*, we first applied a Kruskal–Wallis test in order to identify differences between groups [42]. If the null hypothesis was rejected (*p*-value <0.05), we proceeded with a post hoc analysis (multiple group comparisons) using Dunn’s test and Holm’s correction method [42], i.e., with the aim of identifying those pairs of groups in which such differences occur.

Given the small number of observations in some of the levels of the *Ethnic* variable (e.g., *Asian* (1), *Amerindian* (7)), we decided to redefine this variable using two levels: *Mestizo* and *Non-Mestizo*. The latter level groups observations from the *Asian*, *Amerindian*, *Afro-Ecuadorian*, and *White* levels. In the case of the *Computer at home* variable, we decided to exclude it from our analysis due to the small number of observations (9) within the level labeled as *No*.

Research question 2 was addressed using partial least-squares structural equation modeling (PLS-SEM) [43]. Specifically, we relied on the software *SmartPLS* [44] version 3.3.3. We modeled each construct of the hypothesized model (Figure 1) as a *reflective* latent variable, that is, by assuming that the indicators measured through the administered survey represent the effects or manifestations of an underlying construct.

Although other multivariate approaches are also valid to be applied (e.g., covariance-based SEM (CB-SEM) [45] or multiple regression [30], among others), our choice was based on three main reasons. On the one hand, PLS-SEM has typically been employed to analyze technology acceptance models, especially Moodle [23,27,31,32]. This is mainly due to PLS-SEM’s ability to handle both reflective and formative latent variables. As an additional value of employing this approach, our results could be better compared with those published in the literature. On the other hand, the complexity of our model involving third-order interactions between moderating variables is difficult to address from other approaches (e.g., CB-SEM). Finally, PLS-SEM is less restrictive in relation to the distribution of variables and sample size compared to the CB-SEM approach [43].

A crucial point here is the analysis of moderating effects, which can be performed in PLS-SEM through three approaches [43]: product indicator, orthogonalizing, and two-stage. Following the guidelines provided by Hair et al. [43], we relied on the two-stage approach to model the interaction terms. In this context, variables *Age*, *Gender*, and *Experience* (including their combinations) were modeled as dichotomous, *dummy* variables. Table 3 summarizes how these variables were derived. Note that the original variables (first column) were transformed into dichotomous variables, where 0 corresponds to the reference group and 1 to the other group. Similarly, the rest of the variables resulting from the combination of the first three were transformed into dichotomous variables from the variable levels. Here, it is important to note that the reference group corresponds to when the rest of the groups take values equal to 0. Regardless of the variable, reference groups were not explicitly included in the model, as indicated in Table 3.

In order to assess the measurement model (psychometric properties) of our instrument, we considered both its reliability and validity. Table 4 shows the evaluations made in terms of *convergent validity*, *internal consistency reliability*, and *discriminant Validity*. Note that for each type of assessment, we also included the criteria that have been suggested in the literature [43], to decide whether a value is adequate or not. For example, in the case of *Loadings*, an adequate value is one that is greater than 0.7.

The values presented in Table 4 correspond to the fitting performed in *SmartPLS* using bootstrapping with 5000 subsamples and without including interaction variables. Overall, our instrument shows good psychometric properties. However, we decided to exclude *bi3*, *ee3*, *fc4*, *hm3*, and *si2* indicators from the subsequent structural analyses. The reason was the excessive composite reliability of their respective latent variables [43]. After eliminating these indicators, the criteria of the constructs that were affected were recalculated (as shown in italics in Table 4). It is possible to see that variables *HM* and *SI* maintained a composite reliability slightly greater than 0.95. However, we decided to keep it in our subsequent analyses, as we consider that this excess is marginal. In general, it was observed that the instrument had good psychometric properties. As Furr [39], Bandalos [46], Saltos-Rivas et al. [47] indicate, the quality of the model measurement is a prerequisite for drawing valid and reliable conclusions in subsequent analyses—in our case, through structural models to answer research question 2.

Finally, to assess the structural models, we followed the 6 steps suggested by Hair et al. [43]. These steps assess the model for the following:*Collinearity* issues (VIF criteria);Significance of model relationships ($\beta_{i}$ path coefficients);The level of explained variance ($R^{2}$ and $R_{adjusted}^{2}$);The $f^{2}$ effect size;The predictive relevance ($Q^{2}$);The $q^{2}$ effect size.

The *collinearity* issues WERE assessed in predictor variables through the variance inflation factor (VIF) criteria. We sought to obtain values lower than 5 in order to ensure that *collinearity* was not an issue in the conducted estimation [43]. Regarding the model relationships, we focused on the significance of the path coefficients, that is, in terms of p-values. Here, a p-value below 0.05 indicates that the corresponding effect is significant. From an inferential perspective, this means that the effect is actually different from 0 (no effect) in the population. Furthermore, the relevance of these relationships were interpreted by comparing the direct, indirect, and total effects of the exogenous variables on the endogenous variables. The level of variance explained was analyzed through the coefficient of determination ($R^{2}$) of the endogenous variables. $R^{2}$ ranged from 0 to 1, where values closer to 1 indicate a good predictive power of the model. We also considered it important to include the *adjusted*
$R^{2}$ criterion, which penalizes complex models by taking into account the number of predictive variables. In both cases, we assumed that the values 0.75, 0.50, and 0.25 describe *substantial*, *moderate*, and *weak* predictive powers, respectively [43]. The relative impact of exogenous variables was assessed through the $f^{2}$
*effect size* criterion. The reference values were 0.02 (*small*), 0.15 (*medium*), and 0.35 (*large*) [43]. A value lower than 0.02 meant no effect, that is, the variable in question did not significantly predict the associated exogenous variable. We assessed the predictive relevance of the model through the Stone–Geisser $Q^{2}$ [43]. This measure was computed through a blindfolding procedure, adopting a *cross-validated redundancy* approach with a specific *omission distanceD* ranging from 5 to 10. As recommended by Hair et al. [43], *D* had to be chosen such that it was in the range of 5 to 10 and was not a factor (exact divisor) of the sample size. Since in our case, the sample contained 538 observations, we chose D=7, which represents an omission of 14% of the observations per blindfolding round. An acceptable value for $Q^{2}$ is greater than 0. Finally, the relative impact of predictive relevance was calculated using the $q^{2}$ criterion for each endogenous variable. Similar to $f^{2}$, the values of 0.02, 0.15, and 0.35 were considered as *small*, *medium*, and *large* predictive relevance, respectively [43]. A value below 0.02 meant that the corresponding variable had no relative predictive relevance.

## 5. Results

In this section, we summarize the major findings obtained from the conducted analysis. They were organized according to the research questions formulated in Section 3.

### 5.1. Overall Acceptance Level

As previously established, in order to determine the level of technological acceptance that Ecuadorian university teachers have when it comes to Moodle, we considered the behavior intention construct. Table 5 summarizes the descriptive statistics obtained for each construct considered in the study (n=538). From the values of median and mean, we perceive that, in general, the respondents agreed with most of the items included in the survey. Specifically, we observe that for the *BI* construct, which was intended to capture the users’ acceptance, these values are high (Median=4.667, Mean=4.300, SD=0.858).

In order to obtain a more detailed picture of how the respondents evaluated *BI*, Figure 2 provides the distribution of these evaluations grouped by score and indicator. Clearly, most of the scores were 4 or above, which indicates a high level of acceptance. Specifically, if we consider the proportion of teachers who evaluated the *BI* variable with scores of 4 and 5, the result is approximately 80.3%. According to the indicator labels and this proportion, it is clear that the majority of Ecuadorian teachers have strong intentions to continue using Moodle in their daily work. Based on these pieces of evidence, we can conclude that the level of acceptance of Ecuadorian university professors is high when it comes to Moodle.

### 5.2. Acceptance Level and Demographics

Table 6 shows the results of the variable-level tests performed to detect group-level differences, that is, using the averaged BI as a dependent variable. Evidently, in the case of *Gender* and *Ethnic*, the p-values are greater than 0.05. Therefore, the null hypothesis of equality of medians cannot be rejected. This indicates that Ecuadorian teachers with different gender and ethnicity have very similar levels of acceptance. The results obtained by applying the Kruskal–Wallis test to the rest of the variables indicate that the null hypothesis of equality of medians for the *age group* cannot be rejected. In contrast, for the variables *Education*, *Discipline*, and *Experience*, the p-values were less than 0.05, which indicates the existence of differences at the group level. To detect the pair of groups between which such differences exist, we proceeded with Dunn’s post hoc test, as illustrated in Table 7. Here, the results indicate that for variable *Education*, differences occur between *Bachelor* and *Master* and between *Bachelor* and *Ph.D*. For variable *Discipline*, although the standard p-values (*p*) indicate that there are differences between the *Nat.* group and three others (e.g., *Agri.*, *M&H*, and *Soc.*); the p-values from Holm’s correction method ($p_{holm}$) contradict these results. Finally, for variable *Experience*, we can see that differences occur only in the three comparisons that always include the *High* group. Specifically, *High* is significantly different from *None*, *Low*, and *Moderate*.

In order to better observe these multiple comparisons, Figure 3 shows the medians and 95% confidence intervals for each group. In line with the results of Table 7, the plots in Figure 3 show that while Dunn’s test allows us to identify two groups (e.g., one with *Bachelor* and another with *Master* and *PhD*) in the case of the variable *Education*, this is not possible in the cases of the variables *Discipline* (plot b) and *Experience* (plot c) (e.g., the confidence intervals overlap for each level of the variables).

Another important pattern depicted by Figure 3 is the increasing trend in the acceptance level as *Education* and *Experience* increases. This tell us that the higher the education and previous experience with other LMSs, the greater the professors’ acceptance of Moodle.

In summary, these results reveal that the level of acceptance of Ecuadorian university professors is different according to their education and previous experience with LMSs.

### 5.3. Determinants of the Technological Acceptance

The results of the PLS-SEM analysis on the model hypothesized in Section 3 are summarized in Figure 4. The diagram shows the path coefficients and p-values corresponding to a bootstrapping significance test that assesses whether these coefficients are significantly different from 0. This significance test was performed from 5000 bootstrapping replicates. Note that in these results, we have excluded moderating variables for the moment. The reason behind this decision is to know the extent to which the model without moderator effects is able to explain Moodle acceptance (e.g., through the *BI* variable). The assessment of these results is summarized in Table 8. Here, it is possible to see that there are no *collinearity* issues in the estimated relationships (all VIF values are below 5). Moreover, variable *AS* has significant effects on *PE*, *EE*, *SI*, and *FC*. The path coefficients of these relationships range from 0.502 to 0.768, corresponding to relative impacts ($f^{2}$) ranging from *medium* to *large*. In the case of *BI*, the only variables that significantly explain it are *EE*, *FC*, and *PE*. The corresponding path coefficients are clearly lower here than in the case of *AS*, which range from 0.133 to 0.452. As a result, the relative impacts of these variables on *BI* are categorized as follows: *no effect* (for *PE*) and *small* (for *FC* and *EE*). Although indirect, *AS* is another significant determinant of *BI*. In Table 8, we can see that this total effect, calculated as the sum of the individual indirect effects (e.g., through *FC*, *PE*,*SI*, and *EE*) are equal to 0.614. See also that, consistent with the results obtained for the direct effects, we can see that the path *AS*→*SI*→*BI* is not significant (p>0.05).

Combined, the variables included in this model without the moderating effects explain about 59.4% of the variance of *BI*. However, if the number of exogenous variables included to predict *BI* is taken into account, the value of this explained variance is reduced to 58.8%. In any case, these values are indicative that this model has *moderate* predictive power (e.g., $R^{2},R_{adjusted}^{2}\in \left[0.5,0.75\right]$). A similar result is obtained for the variables *EE* and *FC*, in which cases *AS* explains more than 55.5% of their variance. In contrast, *AS* has a *weak* predictive power for the variables *PE* and *SI* (e.g., $R^{2},R_{adjusted}^{2}\in \left[0.25,0.5\right]$). Finally, the fact that all $Q^{2}$ values are clearly above 0 for the exogenous variables indicates that the model has sufficient predictive relevance. However, from the perspective of the endogenous variables, the $q^{2}$ effect sizes show that only *EE* and *FC* have relative predictive relevance. Specifically, these relative impacts can be regarded as *small* in both cases (e.g., $f^{2}\in \left[0.02,0.15\right]$).

Regarding the moderating effects, Table 9 shows the results corresponding to seven independent models, that is, one for each variable appearing in the columns. These models were derived from the one shown in Figure 4, but including the moderating variables independently. In order to shorten the large number of results obtained for each model, in Table 9, we only report the number of categories that were significant for each variable out of the total number of categories. Since the variables were *dummified* (see Section 4.4), each variable is represented by n−1 categories out of the total *n* that constitute it. Additionally, the reference case (obtained when the *dummy* variables corresponding to the n−1 categories are set to 0) is represented by the direct effect of the relationship to be moderated. Taking these aspects into account, a variable is considered to moderate a given effect when both the coefficients of the n−1
*dummy* variables and the coefficient of the direct effect are significant. Table 9 shows that in no case is this condition achieved. Only partial moderating effects were identified for *Gdr* (on *TV*→*BI*) and *Age&Gdr* (on *FC*→*BI*, *PE*→*BI*, and *TV*→ *BI*). As additional information, we included in Table 9 the predictive power of including the moderating variables in the model without moderating effects (Figure 4). As we can see, in all cases, the predictive power increases, but not enough to change the *moderate* category achieved by the model without moderating effects.

Specific details of these partial effects are listed in Table 10. Note that in the case of the *Gdr* variable, while the *Male* category positively and significantly moderates the *TV*→*BI* relationship, the coefficient associated with the reference case (e.g., *TV*→*BI*) is not significant. Therefore, we conclude that in general, there is not a moderating effect of *Gdr* on *TV*→*BI*. Regarding the *Age&Gdr* variable, Table 10 shows a similar result for the latent variable *TV*. However, for the variables *FC* and *PE*, although the reference cases (e.g., *FC*→*BI* and *PE*→*BI*) have significant effects, this does not occur for some categories included in the model. As a result, we cannot affirm that *Age&Gdr* significantly moderates the relationships predicting *BI*.

## 6. Discussion

In this section, we will discuss the results obtained in our research. First, we will summarize our contributions and emphasize the extent to which they are consistent or not with the experiences reported in the literature. Later, the implications derived from our results will be addressed from both a practitioner and scientific perspective. Finally, an analysis of the main limitations of our research will be made, including some of the future lines of research that could address them.

### 6.1. Summary of Contributions

Our research study addressed three fundamental questions related to the technological acceptance that Ecuadorian university teachers have of Moodle, one of the most widely used LMSs today. The first question focused on the level of technological acceptance. In this regard, our results indicate that the level of acceptance is high. Specifically, more than 80% of respondents agree that they will continue to use Moodle in their daily teaching work. This is in line with the results reported in the meta-analysis conducted by Garcia-Murillo et al. [11]. Compared to the mean proportion estimated by that study (69% with a 95% confidence interval of [59%,78%]), our results are clearly above these values. Other experiences such as [25,27,28,32,37] showed similar results, that is, with average values above the median score of the Likert scale used to measure the constructs associated with acceptance. This means that Ecuadorian university teachers possess levels of acceptance similar to their peers in other regions of the world. Although it is possible that a large part of the teachers who responded to the survey are precisely those who are most satisfied with Moodle, we believe that the high acceptance is largely due to the fact that higher education institutions in Ecuador (which usually teach face-to-face classes) had been using online classes for months at the time the questionnaire was administered. This was due to the COVID-19 pandemic. Thus, much of the learning management at that time occurred through Moodle, which increased its diffusion and use by Ecuadorian teachers.

The second question addressed by our research study delved into whether or not this acceptance is the same for groups of teachers with different demographic characteristics. In this sense, we found that statistically, there are no differences according to the age, gender, or ethnicity of the participants. Similar results were reported by Baytiyeh [28] in the case of gender. In contrast, Costa et al. [25] found differences in terms of both gender and the discipline to which the teachers belong. Although our results agree with Costa et al. [25] when it comes to discipline, the data on which we based our post hoc tests did not allow us to identify the pairs of groups between which such differences occurred. In addition to these demographics studied by previous works, our work found differences in two new characteristics: the academic level (education) of the professors and their previous experience with e-learning systems. In the first case, we identified that teachers whose highest level of education is a university degree accept Moodle at a significantly lower level than those with master’s and PhD degrees. This result suggests that the higher the academic level of the teachers, the stronger their perception of the usefulness of Moodle in their daily work. Regarding previous experience, we found that the teachers who perceive themselves as experts (high level) have a significantly higher acceptance than those who do not. However, the data did not allow us to find differences between those who say they have a *very high* level and the rest of the groups (including the *high* level). Overall, the results suggest a positive association between experience and acceptance, which is in contradiction with previous research [23], where experience was found to be a negative moderator between behavioral intention and use.

Finally, the third question addressed by our study had to do with the identification of determinants of Moodle acceptance. The model taken as a theoretical basis was UTAUT2 [23]. We adapted it to the teaching context by adding two new variables and modifying one of the variables originally proposed by Venkatesh et al. [23]. Overall, our results show that UTAUT2 partially explains behavioral intention (e.g., Moodle acceptance). More specifically, the predictive power of our model without considering moderating effects can be assessed as *moderate*. In that sense, our results are consistent with those reported by Islam [27], Baytiyeh [28], Zwain [32], but not with North-Samardzic and Jiang [31]. Interestingly, the only relationships that were found to be significant in our model were some of those that UTAUT2 inherited from UTAUT (*PE*, *EE*, and *FC*). This leads us to conclude that perhaps the model that best explains acceptance in Ecuadorian teachers is the latter. This fact is in contradiction with the findings reported by North-Samardzic and Jiang [31], where UTAUT was not successful in predicting *BI*; and with Zwain [32], where most of the significant determinants of *BI* were those exclusive to UTAUT2.

An important contribution of our work is that for the first time, attitude strength (*AS*) was considered an (indirect) driver of behavioral intention in the context of university teachers. Specifically, we have partially confirmed the results that Nistor et al. [38] reported in the case of university students that *AS* is a determinant of *PE* and *EE*, which in turn explains *BI*. However, unlike Nistor et al. [38], our results did not indicate that *AS* is a significant determinant of SI. On this particular issue, Nistor et al. [38] found through a multigroup approach that for those students with high levels of *AS*, this construct did not significantly explain SI. So, considering the fact that teachers perceived themselves as having high levels of *AS* in our study, our results are in line with those of Nistor et al. [38]. Another possible explanation for this absence of the IS effect on BI is that by definition, this construct measures the degree to which users (teachers) are sensitive to the opinions of others who are important to them (e.g., peers and managers) regarding Moodle use [22]. Given that the pandemic has caused significant social distancing, it is expected that this construct is not very relevant to teachers’ decisions about whether or not to accept Moodle. As an additional contribution, our study was able to find a significant effect of *AS* on *FC*, a result that Nistor et al. [38] was not able to examine due to problems related to the psychometric properties of FC.

Finally, the fact that we found no moderating effects of participants’ age, gender, or previous experience with LMSs contradicts previously reported experiences [25,31,37] (including the original UTAUT2 study [23]). On this point, we believe that the main cause of this result is the fact that Moodle has now become a *must-have* in higher education institutions in Ecuador. As mentioned above, the current situation has caused most learning to occur online, with the consequent omnipresence of LMSs in educational settings. So, it seems that the old differences that prepandemic studies were able to detect have faded. We recognize, however, that further research is needed to clarify this issue.

To summarize this section, Table 11 lists the contributions on Moodle acceptance in the context of university faculty. We have included several aspects that, in our opinion, are relevant to assess the extent to which our results help advance this line of research. In addition to the results discussed above, it can be seen that our research (last row of Table 11) stands out among the others for being the only one that employs a larger sample of participants and that focuses on South American teachers. It is also easy to observe the high level of heterogeneity of the results reported by the literature. However, this seems to be an issue not only for Moodle, but rather, for LMS-related research in general [48].

### 6.2. Implications

From a practical perspective, our results have important implications. First, the fact that Moodle has high levels of acceptance among Ecuadorian teachers evidences the success of adopting this LMS, especially in these times when online learning has become the main way to promote student learning. Second, the fact that teachers without master’s or doctoral degrees have lower acceptance points to the fact that they require greater attention from policymakers. Similarly, teachers who perceive themselves as having little or no previous experience with LMSs can be part of training courses, with the aim of achieving not only technical competencies with Moodle, but also strengthening their attitudes towards the importance of this type of system in higher education. Finally, the fact that Moodle acceptance is explained by performance expectancy, effort expectancy, and facility conditions gives policymakers the main points to influence in order to achieve higher levels of acceptance. Special attention should be paid to the attitude strength, which is a significant driver of the three direct determinants for predicting acceptance.

Our findings also have valuable research implications. In addition to having shown that South American— and specifically Ecuadorian—teachers accept Moodle in a similar way as their peers in the rest of the world, our results confirm the relevance of UTAUT2 in capturing the drivers of such acceptance. Although this confirmation was partial, that is, without moderator effects and only based on UTAUT constructs, it is undoubtedly a good starting point for obtaining more accurate models. The fact that constructs such as hedonic motivation, habits, social influence, technological innovation, and didactic value were not found to be significant determinants of acceptance is a warning sign that either they require reformulation or university teachers are currently driven by other factors that have not been addressed so far. The experience of Baytiyeh [28], who relied on EFA before moving on to structural analysis, meant that the indicators measured were distributed into factors that are conceptually different from those originally defined in UTAUT. In this way, the author was able to find significant relationships in all the hypotheses tested. This suggests that perhaps such an approach should be employed while adapting technology acceptance models to the educational context.

### 6.3. Limitations and Future Research

Regardless of the importance of the findings of our research, it has some limitations that must be taken into account in order to judge its validity. The first is related to the sampling method used to obtain the data. Although the sample obtained was composed of the minimum number of participants from the institutions that currently use Moodle in Ecuador, the way in which they responded was voluntary. In other words, the sampling was not random. Thus, there is a risk of bias in the results, especially since it is possible that most of the teachers who responded to the survey were the ones that are the most satisfied with Moodle. This limitation is partly compensated for by the sample size, which, although not large enough, is much larger than that used in previously published studies.

Another important limitation is that although the theory of technology acceptance models has been developed around the concept of causality, we cannot claim in our research that such a phenomenon exists, at least not strictly speaking [49]. The main reason is that the results were based on a single cross-sectional measurement (e.g., using an ex post facto design). Thus, it is not possible to isolate the effects of the factors suggested by theory as the main causes of technology acceptance.

Our focus on Ecuadorian university professors, while contributing to the study of a hitherto unexplored population, affects the degree of generalizability of the results obtained. Thus, more research is needed to determine the extent to which these results hold true in the context of other South American countries.

Finally, and precisely because of the cross-sectional data that we have employed, the results obtained here are a snapshot of a reality that is constantly changing. Thus, important questions remain to be answered. Among them, one very interesting question is whether Ecuadorian teachers will continue to accept Moodle in the same way in the case of an eventual return to face-to-face classes. Our future work will be oriented towards addressing these questions and contributing to resolve the limitations of our study.

## 7. Conclusions

Technology acceptance remains a key issue related to the adoption and further use of information systems [50]. In the context of education, and particularly e-learning, LMSs are among the most used software today [51]. This study focused on characterizing and understanding how technological acceptance occurs in the case of Moodle, an open-source LMS with a high presence in higher education all over the world. To this end, we considered a population that has not yet been studied: Ecuadorian university professors. Our results showed that Ecuadorian teachers have high levels of acceptance of Moodle regardless of their age, gender, ethnicity, and discipline. However, this acceptance is significantly higher in teachers with high educational levels and high previous experience with e-learning systems. The main determinants of this acceptance are attitude strength, effort expectancy, performance expectancy, and facilitating conditions.

## Figures and Tables

**Figure 1 behavsci-13-00419-f001:**
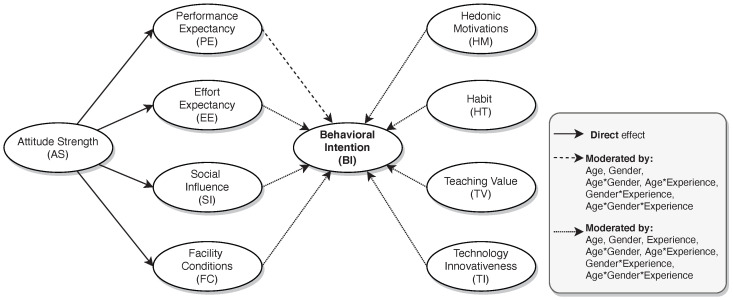
The hypothesized model based on the UTAUT2 model [23].

**Figure 2 behavsci-13-00419-f002:**
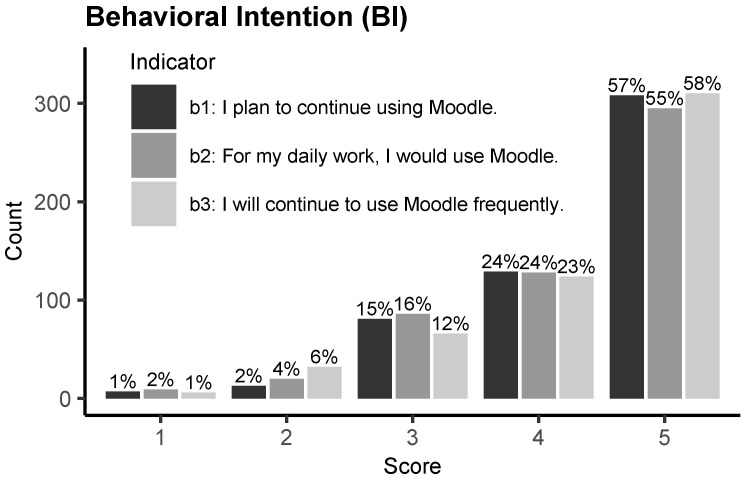
Distribution of the scores per indicator of behavioral intention (*BI*), which is regarded as the technological acceptance of Moodle (n = 538).

**Figure 3 behavsci-13-00419-f003:**
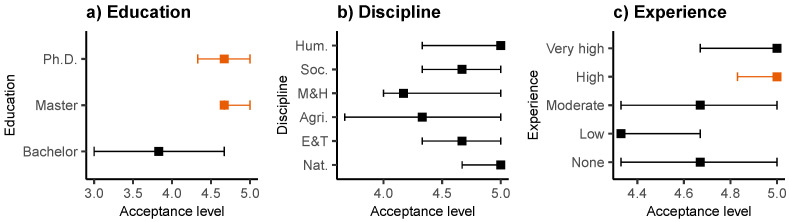
Median (squares) and 95% confidence interval (lines) of acceptance level per group of variables *Education* (**a**), *Discipline* (**b**), and *Experience* (**c**). The statistics were obtained from 5000 bootstrap replicates.

**Figure 4 behavsci-13-00419-f004:**
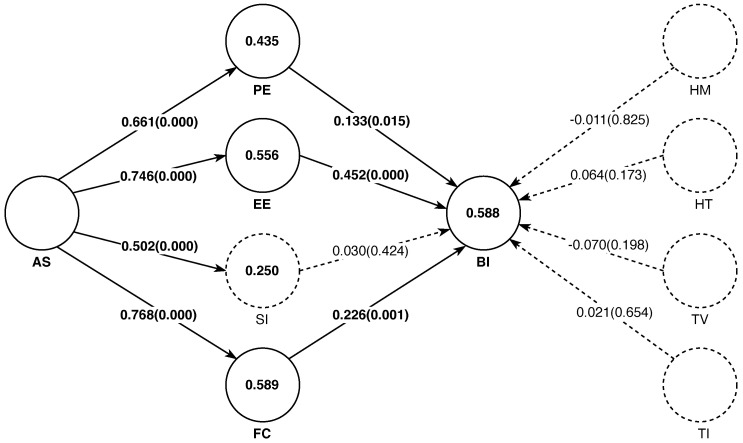
PLS-SEM results for the hypothesized model without moderating effects. The values of the arcs correspond to the path coefficients and p-values (in parentheses) from a two-tailed significance test. Values inside the nodes (circles) correspond to the adjusted coefficient of determination ($R_{adj}^{2}$). All the values shown were obtained from a bias-corrected and accelerated bootstrap method with 5000 replicates.

**Table 1 behavsci-13-00419-t001:** Summary of the tested hypotheses related to the determinants of technological acceptance about Moodle.

Direct/Indirect Effect Hypothesis	Structure	Moderated by (Moderation Hypotheses)						
		Age	Gender	Exp	Age & Gdr	Age & Exp	Gdr & Exp	Age & Gdr & Exp
H1	*AS*→*PE*	–	–	–	–	–	–	–
H2	*AS*→*EE*	–	–	–	–	–	–	–
H3	*AS*→*SI*	–	–	–	–	–	–	–
H4	*AS*→*FC*	–	–	–	–	–	–	–
H5 (indirect)	*AS* →⋯→ *BI*	–	–	–	–	–	–	–
H6	*PE*→*BI*	H6.1	H6.2	H6.3	H6.4	–	–	–
H7	*EE*→*BI*	H7.1	H7.2	H7.3	H7.4	H7.5	H7.6	H7.7
H8	*SI*→*BI*	H8.1	H8.2	H8.3	H8.4	H8.5	H8.6	H8.7
H9	*FC*→*BI*	H9.1	H9.2	H9.3	H9.4	H9.5	H9.6	H9.7
H10	*HM*→*BI*	H10.1	H10.2	H10.3	H10.4	H10.5	H10.6	H10.7
H11	*HT*→*BI*	H11.1	H11.2	H11.3	H11.4	H11.5	H11.6	H11.7
H12	*TV*→*BI*	H12.1	H12.2	H12.3	H12.4	H12.5	H12.6	H12.7
H13	IT → *BI*	H13.1	H13.2	H13.3	H13.4	H13.5	H13.6	H13.7

**Table 2 behavsci-13-00419-t002:** Demographic features of the respondents.

Variable	Level	*n*	%	Cum. %
Age group	Less than 35 years old	82	15.2%	15.2%
	35–44 years	168	31.2%	46.5%
	45–54 years	179	33.3%	79.7%
	55 years and older	109	20.3%	100.0%
Gender	Female	210	39.0%	39.0%
	Male	328	61.0%	100.0%
Ethnic	Afro-Ecuadorian	14	2.6%	2.6%
	Amerindian	6	1.1%	3.7%
	Asian	1	0.2%	3.9%
	Mestizo	470	87.4%	91.3%
	White	47	8.7%	100.0%
Education	Bachelor	40	7.4%	7.4%
	Master	389	72.3%	79.7%
	Ph.D.	109	20.3%	100.0%
Discipline	Natural Sciences (Nat.)	108	20.1%	20.1%
	Engineering and Technology (E&T)	101	18.8%	38.8%
	Agricultural Sciences (Agri.)	27	5.0%	43.9%
	Medical and Health Sciences (M&H)	36	6.7%	50.6%
	Social Sciences (Soc.)	214	39.8%	90.3%
	Humanities (Hum.)	52	9.7%	100.0%
Previous experience with LMS	None	127	23.6%	23.6%
	Low	161	29.9%	53.5%
	Moderate	149	27.7%	81.2%
	High	88	16.4%	97.6%
	Very high	13	2.4%	100.0%
Computer at home	No	8	1.5%	1.5%
	Yes	530	98.5%	100.0%

**Table 3 behavsci-13-00419-t003:** Transformation of moderating variables into *dummy* variables.

Original Variable	Dummy Variable	Value
Age	*Young* (age <45) *	0
	*Old* (age ≥45)	1
Gender	*Female* *	0
	*Male*	1
Experience	*Low* ∈ {None, Low} *	0
	*High* ∈ {Moderate, High, Very High}	1
Age & Gdr	*Young_Female* *	** 0
	*Young_Male*	1
	*Old_Female*	1
	*Old_Male*	1
Age & Exp	*Young_Low* *	** 0
	*Young_High*	1
	*Old_Low*	1
	*Old_High*	1
Gdr and Exp	*Female_Low* *	** 0
	*Female_High*	1
	*Male_Low*	1
	*Male_High*	1
Age & Gdr & Exp	*Young_Female_Low* *	** 0
	*Young_Female_High*	1
	*Young_Male_Low*	1
	*Young_Male_High*	1
	*Old_Female_Low*	1
	*Old_Female_High*	1
	*Old_Male_Low*	1
	*Old_Male_High*	1

Note. * Not explicitly included in the model. ** All other variables must be equal to 0 at the same time.

**Table 4 behavsci-13-00419-t004:** Psychometric assessment of the administered instrument.

Latent Variable	Indicator (Manifested Variable)	Convergent Validity			Internal Consistency Reliability		Discriminant Validity		
		Loadings	Indicator Reliability	AVE	Cronbach’s Alpha	Composite Reliability	Fornell–Larcker Criterion	Cross-Loadings Analysis	HTMT
		>0.7	>0.5	>0.5	[0.60,0.95]	[0.60,0.95]	$\sqrt{\mathrm{AVE}}>$ Correlation with Other Constructs?	Loading > Cross-Loadings with Other Constructs?	Confidence Interval Does Not Include 1?
*AS*	as1	0.854	0.729	0.791	0.867	0.919	Yes	Yes	Yes
	as2	0.906	0.821					Yes	
	as3	0.906	0.821					Yes	
BI	bi1	0.894	0.799	0.866 *0.863* **	0.922 *0.841* **	0.951 *0.926* **	Yes	Yes	Yes
	bi2	0.940	0.884					Yes	
	bi3*	0.957	0.916					Yes	
EE	ee1	0.889	0.790	0.829 *0.859* **	0.931 *0.917* **	0.951 *0.948* **	Yes	Yes	Yes
	ee2	0.930	0.865					Yes	
	ee3	0.924	0.854					Yes	
	ee4*	0.898	0.806					Yes	
FC	fc1	0.830	0.689	0.654 *0.754* **	0.820 *0.837* **	0.882 *0.902* **	Yes	Yes	Yes
	fc2	0.870	0.757					Yes	
	fc3	0.849	0.721					Yes	
	fc4*	0.667	0.445					Yes	
HM	hm1	0.946	0.895	0.890 *0.914* **	0.938 *0.907* **	0.960 *0.955* **	Yes	Yes	Yes
	hm2	0.940	0.884					Yes	
	hm3*	0.943	0.889					Yes	
HT	ht1	0.893	0.797	0.731	0.832	0.890	Yes	Yes	Yes
	ht2	0.820	0.672					Yes	
	ht3	0.850	0.723					Yes	
PE	pe1	0.882	0.778	0.819	0.889	0.931	Yes	Yes	Yes
	pe2	0.940	0.884					Yes	
	pe3	0.893	0.797					Yes	
SI	si1	0.941	0.885	0.893 *0.907* **	0.940 *0.898* **	0.962 *0.951* **	Yes	Yes	Yes
	si2	0.940	0.884					Yes	
	si3 *	0.954	0.910					Yes	
TI	ti1	0.898	0.806	0.759	0.845	0.904	Yes	Yes	Yes
	ti2	0.799	0.638					Yes	
	ti3	0.912	0.832					Yes	
TV	tv1	0.761	0.579	0.758	0.840	0.903	Yes	Yes	Yes
	tv2	0.925	0.856					Yes	
	tv3	0.916	0.839					Yes	

**Table 5 behavsci-13-00419-t005:** Descriptive statistics by latent variable (n = 538).

Latent Variable	Min	Max	Median	Mean	SD
Attitude Strength (*AS*)	1.000	5.000	4.000	3.990	0.839
Behavioral Intention (BI)	1.000	5.000	4.667	4.300	0.885
Effort Expectancy (EE)	1.000	5.000	4.500	4.277	0.822
Facility Conditions (FC)	1.000	5.000	4.250	4.224	0.762
Habit (HT)	1.000	5.000	3.667	3.532	1.049
Hedonic Motivation (HM)	1.000	5.000	4.000	3.861	0.989
Performance Expectancy (PE)	1.000	5.000	4.667	4.289	0.845
Social Influence (SI)	1.000	5.000	4.000	3.687	1.167
Teaching Value (TV)	1.000	5.000	4.000	3.940	0.891
Technology Innovativeness (TI)	1.000	5.000	4.000	3.784	0.958

**Table 6 behavsci-13-00419-t006:** Results from Mann–Whitney U and Kruskal–Wallis tests (n = 538).

Test	Variable	Statistic	df	*p*
Mann–Whitney U	Gender	35,087.000	–	0.697
	Ethnic	16,148.000	–	0.882
Kruskal–Wallis	Age group	2.556	3	0.465
	Education	9.120	2	* 0.010
	Discipline	11.271	5	* 0.046
	Experience	18.798	4	** 0.000

Note. * p<0.05, ** p<0.01.

**Table 7 behavsci-13-00419-t007:** Dunn’s post hoc comparisons (n = 538).

Variable	Comparison	*z*	*p*	$p_{\mathrm{holm}}$
Education	Bachelor–Master	−2.943	** 0.002	** 0.005
	Bachelor–Ph.D.	−2.791	** 0.003	** 0.005
	Master–Ph.D.	−0.252	0.400	0.400
Discipline	Nat.–E&T	1.575	0.058	0.522
	Nat.–Agri.	1.820	* 0.034	0.407
	Nat.–M&H	2.355	** 0.009	0.130
	Nat.–Soc.	2.448	** 0.007	0.108
	Nat.–Hum.	0.039	0.484	1.000
	E&T–Agri.	0.802	0.211	1.000
	E&T–M&H	1.212	0.113	0.862
	E&T–Soc.	0.588	0.278	1.000
	E&T–Hum.	−1.238	0.108	0.862
	Agri.–M&H	0.242	0.404	1.000
	Agri.–Soc.	−0.503	0.308	1.000
	Agri.–Hum.	−1.623	0.052	0.522
	M&H–Soc.	−0.912	0.181	1.000
	M&H–Hum.	−2.060	* 0.020	0.256
	Soc.–Hum.	−1.827	* 0.034	0.407
Experience	None–Low	1.123	0.131	0.466
	None–Moderate	−0.019	0.492	0.732
	None–High	−2.824	** 0.002	* 0.019
	None–Very High	−1.694	* 0.045	0.269
	Low–Moderate	−1.193	0.116	0.466
	Low–High	−3.960	** 0.000	** 0.000
	Low–Very High	−2.173	* 0.015	0.104
	Moderate–High	−2.896	** 0.002	* 0.017
	Moderate–Very High	−1.698	* 0.045	0.269
	High–Very Hh=igh	−0.342	0.366	0.732

Note. * p<0.05, ** p<0.01.

**Table 8 behavsci-13-00419-t008:** PLS-SEM assessment of the hypothesized model without moderating effects. Significance is computed from a bootstrapping method with 5000 replicates.

Endogenous Variable Assessment				
**Path**	**VIF**	**Path Coeff. (β)**	$f^{2}$	$q^{2}$
*AS*→*EE*	1.000	** 0.746	** 1.257	–
*AS*→*FC*	1.000	** 0.768	** 1.435	–
*AS*→*PE*	1.000	** 0.661	** 0.774	–
*AS*→*SI*	1.000	** 0.502	** 0.337	–
*EE*→*BI*	3.819	** 0.452	** 0.132	0.090
*FC*→*BI*	3.987	** 0.226	0.031	0.020
*HM*→*BI*	2.763	−0.011	0.000	−0.006
*HT*→*BI*	2.442	0.064	0.004	−0.002
*PE*→*BI*	3.102	* 0.133	0.014	0.010
*SI*→*BI*	1.850	0.030	0.001	−0.002
*TI*→*BI*	2.676	0.021	0.000	−0.002
*TV*→*BI*	3.206	−0.070	0.004	0.000
Indirect effect assessment				
Path	Total effect			
*AS*→*BI*	** 0.614	–	–	–
*AS*→*EE*→*BI*	** 0.338	–	–	–
*AS*→*FC*→*BI*	** 0.173	–	–	–
*AS*→*PE*→*BI*	* 0.088	–	–	–
*AS*→*SI*→*BI*	0.015	–	–	–
Exogenous variable assessment				
Variable	$R^{2}$	$R_{adjusted}^{2}$	$Q^{2}$	–
*BI*	** 0.594	** 0.588	0.501	–
*EE*	** 0.557	** 0.556	0.474	–
*FC*	** 0.589	** 0.589	0.439	–
*PE*	** 0.436	** 0.435	0.354	–
*SI*	** 0.252	** 0.250	0.224	–

Note. * p<0.05, ** p<0.01.

**Table 9 behavsci-13-00419-t009:** Number of significant categories per moderating variable out of the total of the categories. Significance was obtained from a bootstrapping method with 5000 replicates.

	Moderating Variable						
Path	*Age*	*Gdr*	*Exp*	*Age & Gdr*	*Age & Exp*	*Gdr & Exp*	*Age & Gdr & Exp*
*EE*→*BI*	0/2	0/2	0/2	0/4	0/4	0/4	0/8
*FC*→*BI*	0/2	0/2	0/2	**1/4**	0/4	0/4	0/8
*HM*→*BI*	0/2	0/2	0/2	0/4	0/4	0/4	0/8
*HT*→*BI*	0/2	0/2	0/2	0/4	0/4	0/4	0/8
*PE*→*BI*	0/2	0/2	–	**1/4**	0/4	0/4	0/8
*SI*→*BI*	0/2	0/2	0/2	0/4	0/4	0/4	0/8
*TI*→*BI*	0/2	0/2	–	0/4	0/4	0/4	0/8
*TV*→*BI*	0/2	**1/2**	–	**2/4**	0/4	0/4	0/8
Overall assessment							
$R^{2}$	0.596	0.607	0.609	0.621	0.618	0.624	0.642
$R^{2}$ imp. *	0.2%	1.3%	1.5%	2.7%	2.4%	3%	4.8%
$R_{adj}^{2}$	0.583	0.594	0.599	0.595	0.594	0.600	0.594
$R_{adj}^{2}$ imp. *	−0.5%	0.6%	1.1%	0.7%	0.6%	1.2%	0.6%

Note. * Improvement over the model without moderating effects.

**Table 10 behavsci-13-00419-t010:** The PLS-SEM results for *Gdr* and *Age&Gdr* moderating variables. Significance is computed from a bootstrapping method with 5000 replicates.

Moderator	Latent Variable	Path	Path Coeff. (β).
*Gdr*	*TV*	*TV*→*BI*	−0.054
		*TV*×*Male*→*BI*	** 0.150
*Age & Gdr*	*FC*	*FC*→*BI*	** 0.214
		*FC*×*Old_Female*→*BI*	* 0.182
		*FC*×*Old_Male*→*BI*	0.104
		*FC*×*Young_Male*→*BI*	0.156
	*TV*	*TV*→*BI*	−0.083
		*TV*×*Old_Female*→*BI*	0.079
		*TV*×*Old_Male*→*BI*	* 0.190
		*TV*×*Young_Male*→*BI*	** 0.232
	*PE*	*PE*→*BI*	* 0.135
		*PE*×*Old_Female*→*BI*	* −0.177
		*PE*×*Old_Male*→*BI*	−0.107
		*PE*×*Young_Male*→*BI*	−0.151

Note. * p<0.05, ** p<0.01.

**Table 11 behavsci-13-00419-t011:** Summary of contributions to technological acceptance of Moodle in higher education faculty.

Work	Year	Model	Sample Size	Continent	Acceptance Level	Demographic Differences	Direct Effects Only (without Moderators)			Moderating Effects			Data Analysis Method
							Significant	Not Significant	$R^{2}/R_{\mathrm{adj}}^{2}$	Significant	Not Significant	$R^{2}/R_{\mathrm{adj}}^{2}$	
[27]	2011	From TAM and UTAUT	175	Europe	$\bar{x}=4.530$, s=1.373 (Likert scale 1–7)	Not addressed	PU → Continuance Intention (CI), Access → CI	Compatib. ↛ CI, Perc. behavioral control ↛ CI, SI ↛ CI, PEU ↛ CI	0.702/not reported	Not analyzed	Not analyzed	Not applicable	PLS-SEM
[31]	2015	UTAUT	89	Australia	Not reported	Not addressed	EE → BI	PE ↛ BI, SI ↛ BI	0.360/not reported	GDR × VOL → BI, AGE × VOL → BI, SI × GDR × EXP → BI, GDR × AGE× VOL → BI	AGE, GDR, EXP, VOL, second-order and third-order interact.	0.660/not reported	PLS-SEM, Interaction terms
[28]	2017	From UTAUT	189	Asia	$\bar{x}=3.700$ (Likert scale 1–5)	No differences regarding Gender and Workshop participation	Community influence → BI, Satisfaction → BI, Service quality → BI, Learnability → BI, Technical Quality → BI	None	0.630/ 0.620	Not analyzed	Not analyzed	Not applicable	EFA, Multiple Regression
[25]	2019	TAM	96	Europe	$\bar{x}=4.010$, s=0.843 (Likert scale 1–5)	Difference regarding Gender and Discipline	Not analyzed	Not analyzed	Not analyzed	For Female and Male: SI → PU, PU → ATU, PEU → ATU. For two discipline groups: SI → PU, PU → ATU, PEU → ATU.	SI ↛ PEU	∈[0.684, 0.754]/not reported	Regression, Multigroup approach (sample split)
[32]	2019	UTAUT, TI, IQ, LV	228	Asia	$\bar{x}=3.860$, s=0.950	Not addressed	SI → BI, HM → B I, HT → BI, TI → BI	PE ↛ BI, EE ↛ BI, FC ↛ BI, LV ↛ BI	0.667/ 0.663	Not analyzed	Not analyzed	Not applicable	PLS-SEM
							**Significant**	**Not Significant**	$R^{2}/R_{\mathrm{adj}}^{2}$	**Significant**	**Not Significant**	$R^{2}/R_{\mathrm{adj}}^{2}$	
[37]	2020	TAM	199	Asia	$\bar{x}=3.460$ (Likert scale 1–5)	Not addressed	PU → Satisfaction, PEU $\overset{+}{\to }$ Satisf.	None	Not reported	For Gender, and City PEU → Satisfaction	For Gender, Age, and City PEU ↛ Satisfaction, For Age PU ↛ Satisf.	Not reported	Linear regression, Multigroup approach (sample split)
Our work	2021	UTAUT2 and *AS*, TV, TI	538	South America	$\bar{x}=4.300$, s=0.885 (Likert scale 1–5), 80.3% with scores >3.	No differences for *Age*, *Gender* and *Discipline*. Differences for *Education* and *Experience*.	EE → BI, FC → BI, PE → BI, *AS* → BI (indirect)	HM ↛ BI, HT ↛ BI, SI ↛ BI, TI ↛ BI, TV ↛ BI	0.594/ 0.588	Partial moderation	Age, Gender, Experience, and 2nd- and 3rd-order interactions.	∈[0.596,0.642]/∈[0.583,0.600]	PLS-SEM, Interaction terms from dichotomous and *dummy* variables.

## Data Availability

Data used in this study are available at https://www.doi.org/10.17605/OSF.IO/A4HYK.

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
