# Peer review of "On the Technological Acceptance of Moodle by Higher Education Faculty—A Nationwide Study Based on UTAUT2"

_behavsci, 2023, doi:10.3390/bs13050419_

Round 1

Reviewer 1 Report

This is an interesting paper that aims to contribute to the knowledge about the technological acceptance of Moodle by Ecuadorian university teachers. The paper has theoretical and practical value. The literature review that supports the research is quite well presented. The methodology and the results are robust, and the discussion is aligned with the literature review.

Author Response

We appreciate the reviewer's comments and especially the time and effort spent in reviewing our manuscript.

Reviewer 2 Report

Overall, this is an engaging and interesting paper on the acceptance and adoption of Moodle in Ecuador. This paper is especially important because of the diffusion on online education. The application of the technology acceptance model to better understand teachers' orientation toward model is a good contribution to the field, especially because the authors point out that  teachers are the ones who influence the learning environment. One question I had was whether Moodle was used for online courses exclusively, or if it was used for other courses that have a FTF component. This is addressed in the conclusion, but should be introduced earlier. More clarity here would be useful since that could also impact the acceptance. That said, the results are interesting and show that those with higher levels of education are more likely to adopt. The authors discuss the implications for policymakers, especially in providing more training.

Author Response

The authors are grateful for the reviewer's opinions and suggestion. Regarding the question posed, we have included the following clarification in Section 3 Research Questions and Hypothesized Model:

"Specifically in the context of blended learning, which occurred during the period of social isolation brought on by the Covid-19 pandemic."

Reviewer 3 Report

The paper is well written and methodologically sound. The research questions are clearly presented and the findings are well-founded. Some minor points:

1.    In the abstract, lines 2 and 3, the reference to Moodle should be “its” instead of “their”: “Although its technological acceptance by undergraduate students has been extensively studied in the past, very little is known about its acceptance …”

2.    On page 3, line 137: The SEM acronym should be expanded in its first appearance in the text (Structural Equation Modeling).

3.    On page 18, line 510, the phrase “or work of research found …” could be possibly expressed as “our work found …” or “our research found …”

4.    Also, please note that  Appendix is misplaced on pages 21 and 22. 

Author Response

We appreciate the reviewer's comments and suggestions. In this regard, we have made the suggested corrections (from 1 to 3). However, as for point 4, we cannot make changes because the latex template of the journal controls the order and position of the appendix and tables.

Reviewer 4 Report

Dear Author

Congratulation, a good manuscript indeed.

It is only that; I am a bit confused when the term 'Professors' is used the same as 'Teachers'.

How to distinctive Professors from Teachers in your content of study?

Besides, I don't have an issue with the manuscript as it is presented clearly.

Author Response

We appreciate the reviewer's opinion. Regarding the reviewer's question, indeed, we consider in the context of our research that Teacher and Professor are synonyms. The reason is that the literature consulted uses both terms to refer to university staff.

Reviewer 5 Report

I found this to be an excellent article, pertinent, well written, and scientifically very sound in all aspects, so my recommendation is to be accepted for publication.

I suggest two minor corrections:

- in lines 98-99 (p. 3) there seems to be some confusion, because it indicates reference [25] about Moodle, and relates it to reference [26], about MOOC. Apparently there is some incorrectness here, I suggest that the authors review this aspect.

- in line 276 (p. 8) where is "to the identify" should be "to identify"

Author Response

We appreciate the reviewer's opinions and suggestions. In this regard, we have checked the first point (about references [25] and [26]) and indeed, they are fine: [25] refers to the work of Costa et al. while [26] was included to support the MOOC concept.

Regarding suggested change 2, we have made the correction indicated by the reviewer. That is, we have replaced "to the identify" with "to identify".